# “It Needs a Full-Time Dedicated Person to Do This Job in Our Local Communities with Our Aboriginal Health Services”—Aboriginal and Torres Strait Islander Health Workers and Practitioners Perspectives on Supporting Smoking Cessation during Pregnancy

**DOI:** 10.3390/ijerph20010028

**Published:** 2022-12-20

**Authors:** Amanual Getnet Mersha, Raglan Maddox, Sian Maidment, Kade Booth, Karl Briscoe, Paul Hussein, Hayley Longbottom, Yael Bar-Zeev, Michelle Kennedy

**Affiliations:** 1College of Health, Medicine and Wellbeing, The University of Newcastle, Callaghan, NSW 2308, Australia; 2Equity in Health and Wellbeing Research Program, Hunter Medical Research Institute, The University of Newcastle, New Lambton, NSW 2305, Australia; 3National Centre for Epidemiology and Public Health, The Australian National University, Canberra, ACT 2601, Australia; 4National Association of Aboriginal and Torres Strait Islander Health Workers and Practitioners, Phillip, ACT 2606, Australia; 5Yerin Eleanor Duncan Aboriginal Health Centre, Wyong, NSW 2259, Australia; 6Waminda South Coast Women’s Health and Welfare Aboriginal Corporation, Nowra, NSW 2541, Australia; 7Braun School of Public Health and Community Medicine, Faculty of Medicine, Hebrew University of Jerusalem, Jerusalem 91905, Israel

**Keywords:** aboriginal health practitioners, aboriginal health workers, smoking cessation, smoking in pregnancy, tobacco control

## Abstract

Background: Aboriginal and Torres Strait Islander women deserve improved smoking cessation support. Aboriginal health workers (AHW) and practitioners (AHP) can be central to the provision of culturally safe smoking cessation care (SCC). The objective of this study is to explore attitudes and the perceived role of AHWs/AHPs toward providing SCC to Aboriginal and Torres Strait Islander pregnant women. Method: A mixed-method study using quantitative and qualitative data was conducted among AHW/AHPs in 2021 across Australia. Descriptive and analytical statistics were used to characterise AHWs’/AHPs’ attitudes towards SCC and to evaluate the factors associated with perceptions of who is best placed to provide SCC. Results: From the total AHW/AHP workforce, 21.2% (223) completed the survey. Less than half (48.4%) believed that AHW/AHP were best placed to provide SCC for pregnant women. The majority believed that group-based supports (82.5%) and cultural support programs (63.7%) were the best strategies to support Aboriginal and Torres Strait Islander pregnant women to quit smoking. Conclusion: This study highlights the need to enhance SCC offered to Aboriginal and Torres Strait Islander pregnant women. A targeted workforce dedicated to smoking cessation should be resourced, including funding, standardised training, and ongoing SCC support tailored to Aboriginal and Torres Strait Islander pregnant women.

## 1. Introduction

Australia is recognised as a leader in tobacco control [1]. However, the first peoples of Australia continue to experience inequitable tobacco-related health burdens, which have deep roots in colonial experiences, racism, and dispossession [2]. While a 9.8% reduction in smoking prevalence between 2004–2005 and 2018–2019 was observed among Aboriginal and Torres Strait Islander people [3], half of all the deaths in Aboriginal and Torres Strait Islander people aged 45 and over are caused by smoking [4]. Tobacco use continues to be a leading cause of preventable deaths, and calls have been made for comprehensive approaches to support Aboriginal and Torres Strait Islander people to be smoke-free [5,6,7,8].

There is considerable opportunity to influence the health outcomes for Aboriginal and Torres Strait Islander people. Smoking during pregnancy may be the most compelling “changeable moment”, widely acknowledged as a significant modifiable risk factor for adverse maternal and infant health outcomes [9]. While the past decade has reported improvements in reduced smoking prevalence during pregnancy, in 2020, it was reported that 43% of Aboriginal and Torres Strait Islander women smoked at some stage during their pregnancy [10]. While data often only reports an overall prevalence, recent research led by Kennedy et al. found that Aboriginal and Torres Strait Islander women wanted to quit [11] and that 93% are changing their smoking behaviours, including reducing cigarette consumption during pregnancy [12].

Access to appropriate cessation support, particularly in the primary care setting, is known to increase quitting rates in the general population [13,14]. The Royal Australian College of General Practice Smoking Cessation Guidelines recommends integrating brief advice for all smokers during routine appointments and following up this with a quitting attempt, placing health professionals in a key role in cessation care [15]. Aboriginal and Torres Strait Islander women may receive their maternal health care from a range of health care providers depending on any risks associated with their pregnancy. A previous study of Aboriginal antenatal and postnatal staff confidence and the perceived role and delivery of smoking cessation care to Aboriginal pregnant women found that while most staff assessed women’s smoking status, few offered support via follow-up (28.6%) or provided nicotine replacement therapy (4.7%), and few staff felt confident in motivating Aboriginal and Torres Strait Islander women to quit smoking (19.7%) [16]. However, women from the same service and community reported high acceptance rates when smoking cessation support was offered by the health provider [17]. Aboriginal and Torres Strait Islander health workers have been reported to be best placed to support Aboriginal and Torres Strait Islander women to quit smoking during pregnancy [18]. Aboriginal health workers (AHW) and practitioners (AHP) can be central in providing culturally safe care to Aboriginal and Torres Strait Islander people in both Aboriginal Health Services and mainstream health services in Australia [19].

Traineeship roles are established for AHW/AHP, including Certificate II, III, and IV, Diploma and Advanced Diploma in Aboriginal and/or Torres Strait Islander Primary Health Care (AHW), and Certificate IV and Diploma in Aboriginal and/or Torres Strait Islander Primary Health Care Practice (AHP). There are 1052 AHW/AHP registered nationally with the peak body, The National Aboriginal and Torres Strait Islander Health Workers and Practitioners Association (NAATSIHWP). While AHW/AHP can undertake speciality training, such as mental health, there is currently no AHW/AHP training or qualifications in smoking cessation recognised by NAATSIHWP. In addition, there are no AHW/AHP employed specifically to offer smoking cessation care (SCC) to Aboriginal and Torres Strait Islander women during pregnancy. SCC is frequently added to an AHW/AHP’s role, with AHW/AHP commonly among the first points of contact for Aboriginal and Torres Strait Islander women at a health service. There is a need to understand directly from the workforce their perceived role in SCC, including for Aboriginal and Torres Strait Islander women during pregnancy. This information is important to improve access to appropriate SCC and to improve smoking cessation rates among Aboriginal and Torres Strait Islander people, including during pregnancy. 

This study aims to determine, among the Aboriginal and Torres Strait Islander Health Workers and Practitioners: (a) perceptions of who is best placed to provide SCC for Aboriginal and Torres Strait Islander pregnant women and associated factors; (b) to compare the characteristics of AHW/AHPs based on their attitude to SCC for Aboriginal and Torres Strait Islander women (pregnant and non-pregnant); (c) their role in providing smoking cessation support to Aboriginal and Torres Strait Islander pregnant women; (d) AHW/Ps’ attitude towards various smoking cessation support strategies for Aboriginal and Torres Strait Islander pregnant women.

## 2. Materials and Methods

### 2.1. Research Team

An Indigenous-led research team informed, implemented, and reported this study which is part of a larger project to develop an Indigenous-led evidence base for SCC. We recognise that our lived experiences and worldviews influence the way this work is developed and conducted [20,21,22]. The study was conceptualised and led by MK (Wiradjuri woman) in partnership with NAATSIHWP (KB) and Aboriginal communities represented by: HL, PH. Our team brings Aboriginal and Torres Strait Islander lived experience (MK, KB, HL, SM), Indigenous lived experience (RM), expertise in Aboriginal health services (PH, HL), AHWs (HL, KB), Indigenous tobacco research (AM, MK, RM, KBo, YBZ), and epidemiology (RM, YBZ).

This work was led by the interests and needs of Aboriginal and Torres Strait Islander people and governed by the Which Way? Aboriginal Governance Committee, inclusive of partnering communities [23]. The Which Way? study utilised integrated knowledge translation with a balance of Indigenous knowledge, wisdom, and scientific excellence.

### 2.2. Design and Procedure

A mixed method study with a quantitative and quantitative approach using a cross-sectional online survey was conducted from June to September 2021, targeting AHW and AHP across Australia.

### 2.3. Eligibility Criteria

Participants were eligible if they were: (1) Aboriginal and/or a Torres Strait Islander and (2) AHW or AHP.

### 2.4. Participant Recruitment

Participants were recruited through an email invitation to AHWs and AHPs from June to September 2021. Invitations were sent by the organisation to all members of the NAATSIHWP (*N* = 1052). The survey was also advertised through social media platforms on the National Best Practice Unit, Tackling Indigenous Smoking. Participants could provide contact details to enter a draw for an iPad as an honorarium for their time and expertise at the end of the survey. Data were collected using REDCap software (Version 12.5.5, Nashville, TN, USA).

### 2.5. Survey Instrument

A comprehensive review of the literature informed the survey instrument development in consultation with the Which Way? project partner, AHWs, and NAATSIHWP. The measures used in this survey were co-designed with Aboriginal researchers and community members. This included pilot testing with 10 community members for content acceptability, validity, and reliability prior to data collection. 

*Sociodemographic data*: Sociodemographic variables included Indigenous status; age; gender; smoking status; and state/territory of residence.

*Work-related data*: Participants were asked to indicate whether they were AHW or AHP. The participants’ workplace was categorised as (1) Aboriginal Community Controlled Health Organisation(s) and (2) any other workplace, including general practice. Work experience was collected as a continuous variable and categorised into <5 years, 5–10 years, and >10 years of work experience.

*Smoking cessation training and attitudes*: Participants were asked if they had any kind of smoking cessation training (yes/no). Participants were also asked how comfortable they felt providing smoking cessation support using a five-point Likert scale (not comfortable, somewhat comfortable, unsure, comfortable, very comfortable), which was then re-categorised into a binary yes/no variable: yes, for comfortable and very comfortable, and no for all other answers. The AHW/AHPs’ role in providing SCC for Aboriginal and Torres Strait Islander pregnant women was elicited by using a binary (yes/no) question: *“Does your role currently include smoking cessation support to Aboriginal and/or Torres Strait Islander pregnant women?”*.

To elicit AHW/AHPs’ perceptions of who was best placed to provide SCC for Aboriginal and Torres Strait Islander pregnant women, participants were asked what type of health professional was best placed to provide SCC for (1) pregnant and (2) non-pregnant Aboriginal and Torres Strait Islander people separately. The options included a general practitioner, nurse, midwife, drug and alcohol counsellor, AHWs, AHPs, and others. These were re-categorized to AHW/AHP’s vs. all others.

AHW/Ps’ attitudes towards smoking cessation support strategies for Aboriginal and Torres Strait Islander pregnant women were explored, and participants were asked *Which of the following do you feel is most appropriate for pregnant Aboriginal women to support them to be smoke-free?* Fourteen strategies were included using a binary checkbox (yes/no) response based on previous research with Aboriginal and Torres Strait Islander women [24], such as cultural support, group-based intervention, and Quitline support. An open-text response was also used to elicit an understanding of the AHW/AHP’s role in providing smoking cessation support(s): *“Smoking cessation can take a lot of different forms; where do you see the role of an Aboriginal Health Worker?”*. The qualitative component of the survey allowed for comprehensive insights to address the research aims of who is best placed to provide SCC for Aboriginal and Torres Strait Islander women and the AHW/AHP’s role in providing smoking cessation support. These stories and perspectives from AHW/AHP’s are crucial to gaining health workforce insight and working alongside the qualitative data to provide a ‘bigger picture’ to uncover the best-placed cessation support options for Aboriginal and Torres Strait Islander pregnant women.

### 2.6. Data Analysis

A mixed-method approach was employed to offer a more in-depth understanding of AHW/AHP’s perceptions of their role in the smoking cessation journey. Mixed methods have been recognised as a successful approach to health service research if approached transparently and are justified in addressing the research question [25].

#### 2.6.1. Quantitative Analysis

Descriptive analysis was used to characterise the various variables, which were presented as frequencies and crude percentages. The Chi-square test was used to examine associations between AHW/AHP’s perception of who is best placed to provide SCC for pregnant and non-pregnant Aboriginal and Torres Strait Islander women and socio-demographic variables. Logistic regression was then conducted to assess priori factors (gender, position description ((AHW or AHP)), work experience, smoking status, training, perceived role, and being comfortable in providing SCC) associated with the perceptions of who is best placed to provide SCC for pregnant Aboriginal and Torres Strait Islander women. Factors that were found to have a *p*-value of < 0.2 in the bivariate logistic regression were included in the multivariable logistic regression. An alpha level of 0.05 was used as a cut-off point to identify statistical significance. Findings were presented using the odds ratio (OR) and a 95% confidence interval (CI). Data were analysed in Statistical Package for the Social Sciences (SPSS, Chicago, IL, USA) Version 28.0.

#### 2.6.2. Qualitative Analysis

All analyses of the qualitative data were completed by Aboriginal (SM, MK) and non-Aboriginal (KBo) experienced qualitative health researchers and analysed thematically. The responses were independently coded by SM and KBo using NVivo 12 software to organise the data. Any discrepancies were primarily associated with different wording and terminologies used and were finalised through consultation with MK.

### 2.7. Ethics

As detailed above, this study was developed by an Indigenous-led research team in partnership with Aboriginal and Torres Strait Islander communities. The project upholds Aboriginal and Torres Strait Islander ethical principles, including the National Health and Medical Research Council Guidelines for ethical conduct in Aboriginal and Torres Strait Islander Health Research [26], the Aboriginal Health and Medical Research Council’s Ethical Guidelines: Key Principles (2020) [27], and the international CONSIDER statement [28]. Ethics approvals included the Aboriginal Health and Medical Research Council (#1603/19) and The University of Newcastle (#H-2020-0092). All participants provided informed consent.

## 3. Results

Of 1052 registered NAATSIHWP members, the survey was initiated by 379 participants. Of the 319 who consented, two were deemed ineligible as they were not Aboriginal and/or Torres Strait Islander, with 256 completing the survey. Among the 256 eligible participants, 33 participants did not complete the survey questions aimed at assessing AHW and AHP perceptions of smoking cessation support for Aboriginal and Torres Strait Islander pregnant women and were excluded from this study, providing a final sample of 223 AHW/AHP (21.2% response rate).

### 3.1. Participant Characteristics

The demographic characteristics of the participants are presented in Table 1. Almost half (47.1%, *n* = 105) of the participants were AHWs, and 52.9% were AHPs (*n* = 118). The majority (79.1%) were female, and nearly half (47.1%) were aged between 35 and 54 years, with a mean age of 44 ± 12 years (range 18–72 years old). Over half (57.8%) of participants worked in Aboriginal Community Controlled Health Organisations, and 26.5% reported being a current smoker. More than half (61.9%) had received smoking cessation support training. Three-quarters (74.7%) of participating AHW/AHP felt comfortable providing smoking cessation support (Table 1). The association between training and being comfortable in providing SCC was significant (Chi-squared *p*-value *=* 0.02).

#### 3.1.1. AHW/AHPs’ Perception of Who Is Best Placed to Provide Smoking Cessation Care

A large proportion of AHW/AHP reported that they were best placed to provide support for all Aboriginal and Torres Strait Islander people to quit smoking (*n* = 148; 66.4%). However, this was reduced when reporting pregnancy-specific smoking cessation support, with only 48.4% (*n* = 108) reporting that AHW/AHP were best placed to provide smoking cessation support for pregnant Aboriginal and Torres Strait Islander women. (Figure 1) 

#### 3.1.2. Socio-Demographic Difference of AHW/Ps Based on Their Perception of Who Is Best Placed to Provide SCC for Aboriginal and Torres Strait Islander Women

Approximately one in five (*n* = 47, 21.1%) AHW/AHP believed they were best placed to provide SCC for non-pregnant women but not for pregnant women, and 67 (30%) believed AHW/AHP was not best placed to provide SCC for both pregnant and non-pregnant women (Table 2).

Table 2 shows that a higher proportion of participants in the age group of <35, females, AHPs, work experience between 5 and 10 years, working in general practice, non-smokers, and those with smoking cessation training perceived that AHW/AHP were best placed to provide SCC to both pregnant and non-pregnant women. Higher proportions of AHW/AHP in the following categories males, age group more than 54 years, AHWs, work experience more than 10 years, smokers, and no training reported that AHW/Ps were not best placed to support both pregnant and non-pregnant women (Table 2).

Logistic regression was conducted to evaluate the factors associated with the perception of who is best placed to provide SCC for Aboriginal and Torres Strait Islander pregnant women. In the bivariate logistic regression, a significantly higher proportion of female participants reported that AHW/Ps are best placed to provide SCC for pregnant women (COR = 2.62, 95% CI of 1.31–5.25, *p*-value 0.005). However, this association was not maintained as significant in the multivariate analysis after adjusting for other relevant factors (Table 3).

#### 3.1.3. Qualitative Analysis: AHW/Ps Perceptions of the Role of Aboriginal Health Workers in Smoking Cessation

Two major themes were found: (1)AHW/Ps see themselves as the first point of contact on the quitting smoking journey and as best placed to provide smoking cessation support.

AHW/AHP’s generally recognised their role in supporting Aboriginal and Torres Strait Islander people in their smoking cessation journey. Many reported that they believed that AHW/AHP’s was the first point of contact for people trying to quit.


*“Aboriginal Health Workers should be the first point of contact given the fact they know the community and the people.”*
—Response # 88

They perceived their role as being able to assist in instigating conversations around smoking cessation, which positioned AHW/AHP’s as a key figure in the cessation journey. As emphasised in the quote above, it was common for AHW/AHP’s to note their relationship to the community as a core component of cessation support. Their relationship with people and the community was considered a strength to be able to provide appropriate support by building rapport and trust.


*“At the fore front as they know the community the best”*
—Response #11

The rapport with the community was recognised to allow trust and positioned AHW/AHP’s to form the bridge between people and services. One AHW discussed the context of pregnancy, noting that they are often the first point of contact and can assist mothers with understanding information given to them by midwives. AHW/AHP’s often saw themselves as best-placed to offer support to the mob and as at the forefront of assisting with quitting.


*“We are role models for our mob in our community. I use my story on how I quit smoking and share that its ok to need the support and the support is there to help get them through. Pregnant woman, non-pregnant woman, men, and even teenagers it’s never too late to Quit.”*
—Response #190

The responses suggest that AHW/AHPs tend to assist through emotional support methods coupled with health promotion and education. Many AHW/AHPs suggested that their role was to be “along for the ride” as an advocate and support person through their quit attempt.


*“Front line to support the person’s journey from start to finish.”*
—Response #65


*“Aboriginal Health Workers are the first point of call for our people in the community, they have already formed a trust with our people, and our client will often open up to them if they are struggling with anything”*
—Response #52

They expressed the need to be “supportive”, “non-judgmental”, and “encouraging”, with one participant stating that they felt that the duty of an AHW/AHP in smoking cessation care is to be a “role model” for the community. These responses suggest that through connections with the community, AHW/AHPs take it upon themselves to offer emotional support and advocacy in people’s smoking cessation journeys.


*“I see the role of an Aboriginal Health Worker as a position to support and motivate patients seeking smoking cessation. It’s important to have that constant support when trying to quit. Especially for our young mob who so often give into peer pressure”*
—Response #112

It was common for AHW/AHPs to suggest that their role is to offer health promotion through appropriate education. Through their own positioning, they acknowledged their ability to translate health information and referral to appropriate services.


*“AHWs are at the frontline in terms of communicating with community members. AHW have deep understanding as to why people smoke: triggers, social barriers, cultural reasons, and sometimes lived experiences from the practitioners can also be valuable insights into assisting others and be helpful motivators to their community….”she/he done it, surely I can too” Setting positive lifestyle examples on the frontline.”*
—Response #3

The AHW/AHP’s recognised their expertise and knowledge of risks associated with smoking, as well as the means to appropriately relay the information to the community. Some indicated that they believed it was important to have conversations about smoking cessation during every visit.


*“I think smoking cessation should be in everyday conversation with our client and take every opportunity to have these conversations to ensure our client knows what their options are and how to prevent their health from worsening.”*
—Response #20


(2)AHW/Ps recognise barriers to providing smoking cessation support.


Despite a clear desire to want to help people quit smoking, AHW/AHP’s expressed several barriers regarding the level of assistance they could offer. Time was considered a major factor in the ability to provide suitable cessation care during routine practice. AHWs/AHPs are usually occupied with multiple responsibilities, including but not limited to cultural support/advocacy, patient triage, health promotion/education, community support, making referrals, providing cultural supervision to staff, and supporting the community with health appointments. 


*The Aboriginal Health Worker can be a big part in cessation, but first, the service needs to have the health worker staff to do it. We screen, we have no time to do anything else.*
—Response #36

It was recognised that while AHW/AHP’s can play a significant part in helping people quit, due to the extensive nature of AHW/AHP’s workload, they do not necessarily have the capacity to do so. It was noted that while AHW/AHP’s were happy to share advice with the community, they often referred to trained professionals in smoking cessation. However, it was mentioned that women might be less likely to make the next step due to the lack of relationships established with other professionals.


*“Culturally supporting our women when they are receiving the advice from a trained profession who is giving the advice. I can have a good chat with my woman who ask for my advice, but I always refer to the trained professionals. It is very rare when my clients want this appointment or advice from someone who they are not familiar/comfortable with, so I will support them, which also gives me a good opportunity to put some plans in place to reach the goals clients want to.”*
—Response #109

One AHW described that while they often spoke to women about the risks associated with smoking during pregnancy for their baby, they also highlighted that they were not in a position to administer therapy or a referral.


*“This role is there to provide guidance and a referral pathway to women when they are facing doubt about quitting. AHW can support and educate the women and family on benefits and health risks that smoking may cause to their baby, but it cannot administer any form of therapy other an advice and referral.”*
—Response #105

Those who mentioned time and resources as a barrier to providing smoking cessation care to the capacity that they would like often suggested that a person dedicated specifically to that role in services could help bridge the gap.


*“We have many roles as an AHW/AHP and don’t have enough time to provide the full support, education, and resources to deal with smoking or helping clients reduce or stop smoking. It needs a full-time dedicated person to do this job in our local communities with our Aboriginal Health Services.”*
—Response #99

#### 3.1.4. AHW/AHPs Recommendations on Smoking Cessation Support Strategies for Aboriginal and Torres Strait Islander Pregnant Women

Eight out of ten AHW/AHPs (82.5%) believe group-based smoking cessation supports are the best strategy to support Aboriginal and Torres Strait Islander pregnant women quitting smoking (Figure 2). Programs that incorporate culture and cultural practices were reported by 63.7% of the AHW/AHP as the best strategy to support Aboriginal and Torres Strait Islander pregnant women quit smoking. Aboriginal and Torres Strait Islander communities are diverse, and as such, the ways in which culture and cultural practices incorporate would be driven by the community and vary across the country.

One-on-one counselling was selected as a preferred method for supporting Aboriginal and Torres Strait Islander pregnant women by 42.1% of the participating AHW/AHP. Self-directed support, including Quitline and mobile phone apps, was selected by 35.4% of the participants.

Holistic support, such as acupuncture, art, bush medicine, exercise, and yoga, was reported by 36.8% of AHW/AHPs as a potential smoking cessation support strategy for Indigenous pregnant women. Holistic views on improving health and wellbeing emphasise the connectedness of the physical, social, emotional, cultural, and spiritual well-being of both the individual and the community. (Figure 2)

## 4. Discussion

Aboriginal and Torres Strait Islander women who smoke are likely to attempt to quit smoking during pregnancy [11,12,17]. While two-thirds (66.4%) of AHW/AHP believed they were best placed to provide smoking cessation support for Aboriginal and Torres Strait Islander people, less than half (48.4%) reported this same role for pregnant women. This is the first national study to explore the perceived role of AHW/AHP in providing SCC for Aboriginal and Torres Strait Islander people, including during pregnancy. 

Pregnancy provides an opportunity to address and mitigate the harms of smoking and support cessation. As such, Aboriginal and Torres Strait Islander women have the right to culturally, safely, and responsively quit smoking supports. Training is reported to improve the confidence of healthcare providers when providing SCC [29]. An evaluation of the effectiveness of smoking cessation training tailored towards health professionals working with Aboriginal and Torres Strait Islander people found significant improvements in health professionals’ confidence, skills, and knowledge to provide SCC for Aboriginal and Torres Strait Islander people. Training included information about the health consequences of smoking and smoking cessation methods, such as brief intervention techniques [30].

Aboriginal and Torres Strait Islander women have reported the need for improved support from their healthcare providers to successfully quit smoking during pregnancy [31]. Previous research has identified the potential improvement of SCC for Aboriginal and Torres Strait Islander pregnant women through the provision of support by AHW/AHP [18]. However, to date, no one has reported AHW/AHP perspectives on this role. Calls for AHW/AHP to provide SCC to Aboriginal and Torres Strait Islander women are in part due to the low confidence reported in the general maternal health workforce [16,32]. Less than half (48.4%) of the AHW/AHPs in our study indicated that they felt they were best placed to provide SCC for Aboriginal and Torres Strait Islander pregnant women. Respectively, 5.8%, 30.9%, and 14.8% of AHW/AHPs believed that general practitioners, nurses, midwives, and other health professionals, including multidisciplinary teams, were best placed to provide SCC care for Aboriginal and Torres Strait Islander pregnant women. These findings indicate there is no consensus on who is currently best placed to provide SCC to pregnant Aboriginal and Torres Strait Islander women. As such, it is likely that Aboriginal and Torres Strait Islander women are not receiving appropriate guidance and support to successfully quit smoking during pregnancy. 

Previous qualitative research reported a reluctance among AHW/AHP to discuss smoking cessation with pregnant women due to fear of making the women feel bad and the potential to damage health provider–patient relationships [33]. This was also reported among a range of maternal health care providers who reported fear of pushing women away from their antenatal care follow-ups by advising about smoking cessation [34]. Our study did not find this; in fact, AHW/AHPs most frequently reported the significance of AHW/AHPs in having conversations about quitting with their community. However, the time required for SCC and the competing demands on AHW/AHP roles were noted as barriers to providing smoking cessation care. AHW/AHPs suggest that a specific SCC role in an Aboriginal Health Service would improve SCC offered to Aboriginal and Torres Strait Islander people, including during pregnancy. 

In Australia, telephone-based counselling is offered free of charge through the Quitline service [35]. While AHW/AHPs can refer clients to the Quitline, or clients can self-refer, our study found that AHW/AHPs report a reluctance in Aboriginal and Torres Strait Islander people to use such supports as they do not have an established relationship. AHW/AHPs reported group-based smoking cessation supports to be most appropriate for Aboriginal and Torres Strait Islander women during pregnancy. This is in line with what Aboriginal and Torres Strait Islander women have reported as meaningful SCC care [24]. While group-based smoking cessation support is reported as effective and offered internationally [36], in Australia, there is no funded smoking cessation group-based support for health workers to refer clients to. Broadening SCC beyond frontline health professionals can help provide comprehensive support and ease the burden of responsibility for time-constrained AHW/AHP.

There is a critical need to support Aboriginal and Torres Strait Islander women to quit smoking during pregnancy. The high prevalence rates, despite a high motivation to quit smoking among Aboriginal and Torres Strait Islander pregnant women, suggests more could be conducted to increase the rate of smoke-free pregnancies [12]. This is consistent with the draft National Tobacco Strategy [37] and the WHO Framework Convention on Tobacco Control [38], which report that providing comprehensive tobacco control approaches, including high-quality, culturally safe smoking cessation care, can support significant reductions in smoking prevalence. Evidence from this study suggests this should include a dedicated smoking cessation workforce, which could actively assist in supporting Aboriginal and Torres Strait Islander women, pregnant and non-pregnant, to quit and stay smoke-free. 

## 5. Conclusions

AHW/AHPs were recognised as well-placed to provide culturally safe and responsive smoking cessation care. AHW/AHPs relationship and knowledge of the community is beneficial in relaying information and health promotion and going beyond, providing guidance to becoming a part of the quitting journey as a mentor and advocate. While most AHW/AHPs feel that they are well positioned to support Aboriginal and Torres Strait Islander people through their quitting journey, less than half report this during pregnancy. Despite the desire to assist in supporting community members to quit smoking, barriers remain to providing routine smoking cessation care, such as a lack of training, the appropriateness of external referrals, and the capacity of workloads. Comprehensive smoking cessation care could and should be offered to Aboriginal and Torres Strait Islander women during pregnancy. However, this requires appropriate funding, resourcing, and the implementation of a skilled smoking cessation workforce in Aboriginal health services.

## Figures and Tables

**Figure 1 ijerph-20-00028-f001:**
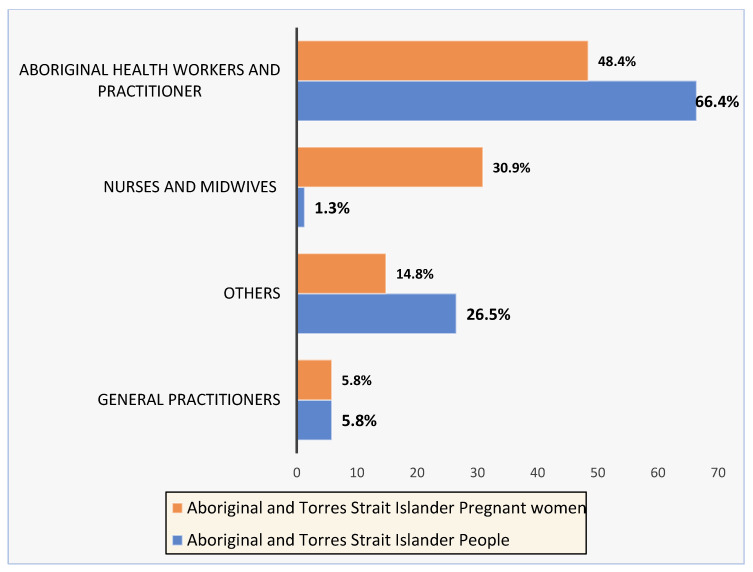
AHW/Ps’ perception of who is best placed to provide SCC to pregnant and non-pregnant Aboriginal and Torres Strait Islander people.

**Figure 2 ijerph-20-00028-f002:**
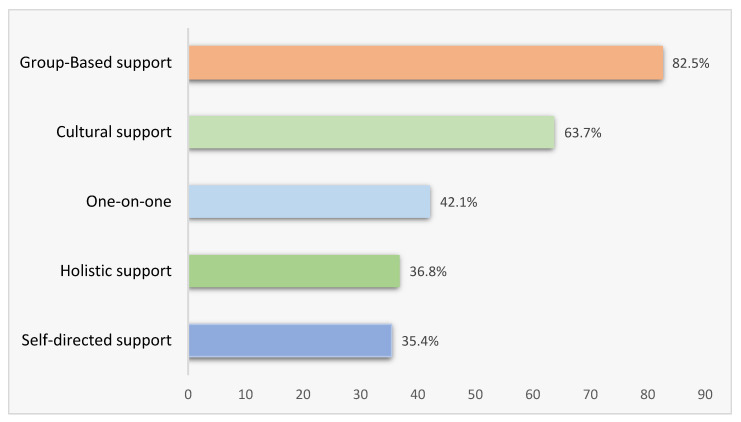
Aboriginal health workers and practitioners’ smoking support strategy preference for pregnant Indigenous women.

**Table 1 ijerph-20-00028-t001:** The characteristics of Aboriginal Health Workers and Practitioners participants (*N* = 223).

Variables	Frequency (*n*, %)
Indigenous status	
Aboriginal	195 (87.4%)
Torres Strait Islander	16 (7.2%)
Aboriginal and Torres Strait Islander	12 (5.4%)
Age (mean ± SD)	44.58 ± 12.4
<35 years old	61 (27.4%)
35–54 years old	105 (47.1%)
>54 years old	57 (25.6%)
Gender	
Men	46 (20.9%)
Women	174 (79.1%)
State	
New South Wales	100 (44.8%)
Victoria	19 (8.5%)
Queensland	56 (25.1%)
South Australia	21 (9.4%)
Western Australia	15 (6.7%)
Tasmania	3 (1.3%)
Northern Territory	9 (4.0%)
Position	
Aboriginal health worker	105 (47.1%)
Aboriginal health practitioner	118 (52.9%)
Professional work experience	
Less than 5 years	58 (26.0%)
5–10 years	60 (26.9%)
10+ years	105 (47.1%)
Workplace	
Aboriginal Community Controlled Health Organisations	129 (57.8%)
General Practice	94 (42.2%)
Smoking status	
Non-smoker	164 (73.5%)
Smoker	59 (26.5%)
Smoking cessation training	
Yes	138 (61.9%)
No	85 (38.1%)
Feel comfortable to provide SCC	
Yes	124 (74.7%)
No	42 (25.3%)

Crude percentages (%) may not add to 100% due to rounding. SCC—smoking cessation care.

**Table 2 ijerph-20-00028-t002:** Socio-demographic variations based on perception of who is best placed to provide SCC for pregnant and non-pregnant Indigenous women.

Variables	Believe AHW/P Are Best Placed to Provide SCC for Both Pregnant and Non-Pregnant Women (*n* = 100)	Believe AHW/P Are Best Placed to Provide SCC for Non-Pregnant Women but Not for Pregnant Women (*n* = 47)	Believe AHW/P Are Best Placed to Provide SCC for Pregnant Women but Not for Non-Pregnant Women (*n* = 9)	Believe AHW/P Are Not Best Placed to Provide SCC for Pregnant and Non-Pregnant Women (*n* = 67)	Chi-Squared *p*-Value
**Age**					0.168
<35 years old	29 (47.5%)	19 (31.1%)	2 (3.3%)	11 (18%)	
35–54 years old	47 (44.8%)	19 (18.1%)	5 (4.8%)	34 (32.4%)	
>54 years old	24 (42.1%)	9 (15.8%)	2 (3.5%)	22 (38.6%)	
**Gender**					0.040 *
Male	14 (30.4%)	16 (34.8%)	1 (2.2%)	15 (32.6%)	
Female	86 (49.4%)	31 (17.8%)	7 (4%)	50 (28.7%)	
**Position**					0.008 *
Aboriginal Health Worker	45 (42.9%)	15 (14.3%)	3 (2.9%)	42 (40%)	
Aboriginal Health Practitioner	55 (46.6%)	32 (27.1%)	6 (5.1%)	25 (21.2%)	
**Work experience**					0.107
Less than 5 years	24 (41.4%)	17 (29.3%)	1 (1.7%)	16 (27.6%)	
5–10 years	34 (56.7%)	11 (18.3%)	1 (1.7%)	14 (23.3%)	
10+ years	42 (40%)	19 (18.1%)	7 (6.7%)	37 (35.2%)	
**Workplace**					0.220
Aboriginal Community Controlled Health Organisations	56 (43.4%)	32 (24.8%)	3 (2.3%)	38 (29.5%)	
General Practice	44 (46.8%)	15 (16%)	6 (6.4%)	29 (30.9%)	
**Smoking status**					0.267
Non-smoker	76 (46.3%)	38 (23.2%)	6 (3.7%)	44 (26.8%)	
Smoker	24 (40.7%)	9 (15.3%)	3 (5.1%)	23 (39%)	
**Training**					0.438
Yes	66 (47.8%)	30 (21.7%)	4 (2.9%)	38 (27.5%)	
No	34 (40%)	17 (20%)	5 (5.9%)	29 (34.1%)	

* Statistically significant (*p* < 0.05).

**Table 3 ijerph-20-00028-t003:** Factors associated with professional perceived role and perception of who is best placed to provide SCC for pregnant Indigenous women (*N* = 223).

Variables	Best Placed Healthcare Provider to Provide SCC			
	AHW/AHPs	Others	COR (95% CI)	AOR (95% CI)
**Gender**				
Male	14 (30.4%)	32 (69.6%)	Ref	Ref
Female	93 (53.4%)	81 (46.6%)	2.62 (1.31–5.25) *	2.09 (0.92–4.74)
**Position**				
Aboriginal health workers	48 (45.7%)	57 (54.3%)	Ref	—
Aboriginal health practitioners	60 (50.8%)	58 (49.2%)	1.22 (0.72–2.08)	—
**Professional work experience**				
Less than 5 years	25(43.1%)	33(56.9%)	Ref	Ref
5–10 years	34(56.7%)	26(43.3%)	1.72(0.83–3.57) ¶	1.94(0.84–4.45)
10+ years	49(46.7%)	56(53.3%)	1.15(0.60–2.20)	1.18(0.53–2.61)
**Workplace**				
Aboriginal Community Controlled Health Organisations	58(45.0%)	71(55.0%)	0.71(0.42–1.22) ¶	0.87(0.44–1.69)
General Practice	50(53.2%)	44(46.8%)	Ref	Ref
**Smoking status**				
Non-smoker	82(50.0%)	82(50.0%)	1.26(0.69–2.30)	—
Smoker	26(44.1%)	33(55.9%)	Ref	—
**Training in smoking cessation support**				
Yes	69(50.0%)	69(50.0%)	1.17(0.68–2.02)	1.30(0.62–2.74)
No	39(45.9%)	46(54.1%)	Ref	Ref
**Feel comfortable to provide smoking cessation support for pregnant women**				
Yes	62(50.0%)	62(50.0%)	1.47(0.72–2.99)	1.76(0.81–3.80)
No	17(40.5%)	25(59.5%)	Ref	Ref
**Perceived Role**				
Yes	70(47.3%	78(52.7%)	0.87(0.50–1.52)	0.57(0.18–1.76)
No	38(50.7%)	37(49.3%)	Ref	Ref

¶ indicates a *p*-value < 0.2 in the univariate analysis and is included in the multivariate logistic regression; * *p*-value < 0.05; COR—crude odds ratio; AOR—adjusted odds ratio.

## Data Availability

All relevant materials and data supporting the findings of this study are contained within the manuscript.

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
