# Peer review of "“It Needs a Full-Time Dedicated Person to Do This Job in Our Local Communities with Our Aboriginal Health Services”—Aboriginal and Torres Strait Islander Health Workers and Practitioners Perspectives on Supporting Smoking Cessation during Pregnancy"

_ijerph, 2022, doi:10.3390/ijerph20010028_

Round 1
Reviewer 1 Report
Very nicely written report on an important topic. My only minor suggestion would be to minimize use of non-standard abbreviations as it makes the manuscript a bit more difficulty to read for the uninitiated.
PLEASE NOTE: These comments are for editor and authors ONLY. YOU DO NOT have my permission to publish these comments.
Author Response
Please see the attachment.
Thank you for this suggestion.
Abbreviations used less frequently in the manuscript such as RACGP, AHS, and AH &MRC are now expanded.
We have also included used abbreviations in the ‘’abbreviation section’’ of the manuscript.
Page number: 14 – 15, Line number: 470-476

Reviewer 2 Report
I commend the authors for doing this study. Policy makers often will make assumptions like AHWs/AHPs must be best placed to offer smoking cessation to pregnant/non pregnant clients, but the authors take the important step of asking the AHWs/AHPs themselves.
I think a mixed- method is most suited to this type of study, and including specific comments really enhances the paper.
Suggestions:
1) Figure 2 needs more detailed explanation. Although you have included a few words of explanation, please tell the reader in more detail what does it mean to the AHW/AHP when you say "cultural" or "holistic"
This specific section could be re written to include more detail re above terms.
"Eight out of ten AHW/AHP (82.5%) believe group-based smoking cessation supports 361 is the best strategy to support Aboriginal and Torres Strait Islander pregnant women quit- 362 ting smoking (Figure 2). Cultural support programs were reported by 63.7% of the 363 AHW/AHP as the best strategy to support Aboriginal and Torres Strait Islander pregnant 364 women quit smoking. One-on-one counselling was selected as a preferred method of sup- 365 porting Aboriginal and Torres Strait Islander pregnant women by 42.1% of the participat- 366 ing AHW/AHP. Self-directed support including Quitline, and mobile phone apps were 367 selected by 35.4% of the participants. Holistic support such as Bush medicine, Yoga, and 368 Acupuncture was reported by 36.8% of AHW/AHP as a potential smoking cessation sup- 369 port strategy for Indigenous pregnant women"
2) Some survey responders point out lack of time as a reason for feeling they are not best suited to this role. You have an international readership, and all are not familiar with the challenges of indigenous people. Maybe tell the reader briefly what other kind of work the AHW/AHPs do.
3) I think adding a quote in your title is not a good strategy. (“It needs a full-time dedicated person to do this job in our local communities with our Aboriginal Health Services)”
You as researchers are reporting findings. Several responders did say they feel they are best placed for the job. Picking a quote as title makes your paper appear biased. Suggest leave it out and keep the rest of the title, which accurately reflects what the paper is about. Ok to include quote in your discussion section, but as part of the title, even though I understand you are trying to make a point, I think it detracts from your overall quality of paper.
Otherwise, my congratulations and deep appreciation for working with indigenous populations and trying to understand their health workers' needs. I enjoyed reading your paper, and hope such work continues in all parts of the world.
